# Assessing the Role of Ancestral Fragments and Selection Signatures by Whole-Genome Scanning in Dehong Humped Cattle at the China–Myanmar Border

**DOI:** 10.3390/biology11091331

**Published:** 2022-09-09

**Authors:** Xiaoyu Luo, Shuang Li, Yingran Liu, Zulfiqar Ahmed, Fuwen Wang, Jianyong Liu, Jicai Zhang, Ningbo Chen, Chuzhao Lei, Bizhi Huang

**Affiliations:** 1Yunnan Academy of Grassland and Animal Science, Kunming 650212, China; 2Key Laboratory of Animal Genetics, Breeding and Reproduction of Shaanxi Province, College of Animal Science and Technology, Northwest A&F University, Yangling 712100, China; 3Faculty of Veterinary and Animal Sciences, University of Poonch, Rawalakot 1235, Pakistan

**Keywords:** Whole-genome resequencing, Dehong humped cattle, *ABHD6*, selection signatures

## Abstract

**Simple Summary:**

We provide a comprehensive overview of sequence variations in Dehong humped cattle genomes with important implications for the future; moreover, we unravel the characteristics of other native cattle in China. In the current study, we investigated a high level of genomic diversities using whole-genome resequencing data in 18 Dehong humped cattle. It is speculated that Dehong humped cattle were influenced by two ancestral segments (Chinese indicine and Indian indicine cattle). Additionally, the selection signatures were detected in genomic regions that are possibly related to economically important traits in Dehong humped cattle. These results will establish a foundation for conservation and breeding programs in the future.

**Abstract:**

Dehong humped cattle are precious livestock resources of Yunnan Province, China; they have typical zebu traits. Here, we investigated their genetic characteristics using whole-genome resequencing data of Dehong humped animals (n = 18). When comparing our data with the publicly-available data, we found that Dehong humped cattle have high nucleotide diversity. Based on clustering models in a population structure analysis, Dehong humped cattle had a mutual genome ancestor with Chinese and Indian indicine cattle. While using the RFMix method, it is speculated that the body sizes of Dehong humped cattle were influenced by the Chinese indicine segments and that the immune systems of Dehong humped cattle were affected by additional ancestral segments (Indian indicine). Furthermore, we explored the position selection regions harboring genes in the Dehong humped cattle, which were related to heat tolerance (*FILIP1L*, *ABHD6*) and immune responses (*GZMM*, *PRKCZ*, *STOML2*, *LRBA*, *PIK3CD*). Notably, missense mutations were detected in the candidate gene *ABHD6* (c.870C>A p.Asp290Glu; c.987C>A p.Ser329Arg). The missense mutations may have implications for Dehong humped cattle adaptation to hot environments. This study provides valuable genomic resource data at the genome-wide level and paves the way for future genetic breeding work in the Dehong humped cattle.

## 1. Introduction

Domestic cattle can diverge into two subspecies: the humpless taurine (*Bos taurus*) and the humped indicine (*Bos indicus*) [1]. They play important roles in societies and economies around the world. India is the second largest producer of world cattle, with 194 million heads of cattle, which mainly belong to indicine. The Indus Valley in India–Pakistan was likely the center of *B. indicus* during the cattle domestication period [2]. Later, *B. indicus* might have been dispersed into East Asia; one important infiltration method to get into China was through Yunnan Province [3]. Therefore, it was the reason for the development of the genetic diversity of cattle breeds in Yunnan Province.

Dehong humped cattle, bearing typical zebu traits, are precious livestock resources of Yunnan Province. This breed has distinctive characteristics (the hump and pendulous skin). It is adapted to harsh conditions and hot climates. Moreover, 300 years ago, the “Gala” cattle (type of zebu cattle) were introduced through the Myanmar tract and reached the Dehong Autonomous Prefecture in the western Yunnan Province of China. Subsequently, Dehong humped cattle were produced as a result of a cross between the zebu cattle and local yellow cattle [4]. The Dehong Autonomous Prefecture, where Dehong humped cattle are located, is surrounded by Myanmar from three sides and separated by two rivers that form a natural geographical barrier. In recent years, most local breeds in China have been ‘crossed’ with foreign cattle to improve the quality. Due to the special geographical environment, they are basically free from the infections of foreign commercial cattle. Therefore, Dehong humped cattle have relatively stable genetic characteristics.

Previous studies have used Illumina BovineHD BeadChip (777K) for the analysis of Dehong humped cattle, indicating that Dehong humped cattle are pure Zebu cattle [4,5]. One study showed that Dehong humped cattle had the lowest heterozygosity in the Yunnan Province, and another other study used statistical methods to detect genes associated with heat tolerance and immunity. In fact, such methods also have limitations, as it is difficult to detect important genetic information due to only known sequences.

Here, we generated the genome resequencing data of 18 Dehong humped cattle and then compared them with reference cattle from different geographical regions in East Asia (India–Pakistan, Korea, and North and South China) [6]. The current data are breakthroughs in the genetic diversities and the candidate signatures of positive selections and in tracing the ancestry components of the Dehong humped cattle.

## 2. Materials and Methods

### 2.1. Ethics Statement

The study was approved by the Institutional Animal Care and Use Committee of Northwest A&F University (2011-31,101,684), following the recommendations by the Regulations for the Administration of Affairs Concerning Experimental Animals of China. Specific consent procedures were not required for this study following the recommendation of the Regulations for the Administration of Affairs Concerning Experimental Animals of China. All operations and experimental procedures complied with the National Standard of Laboratory Animal Guidelines for Ethical Review of Animal Welfare (GB/T 35892-2018) and the Guide for the Care and Use of Laboratory Animals: Eighth Edition.

### 2.2. Sampling

A total of 18 Dehong humped cattle were sampled in China. These samples were collected in Yunnan Province, China. Ear tissues were collected and preserved in 96% ethanol for one day until DNA extraction.

### 2.3. Production of WGS Data

Whole genomes were re-sequenced via Illumina NovaSeq 6000 with 2 × 150 bp models at Novogene Bioinformatics Institute, Beijing, China; 150 bp paired-end sequence data were generated. Additionally, whole-genome resequencing (WGS) samples at 12× coverage, representing 4 breeds, were provided by the NCBI sequence Read Archive, including 19 Chinese indicine cattle (Wannan—5, Guangfeng—4, Ji’an—4, Leiqiong—3, Jinjiang—3), 10 India–Pakistan cattle (Brahman—4, Tharparkar—1, Hariana—1, Nelore—1, Gir—2, and unknown-1), 9—Yanbian, and 15—Hanwoo. In total, 79 whole genomes of cattle were used for the subsequent analysis.

### 2.4. Identification of Single Nucleotide Polymorphisms from WGS Data

At first, the clean reads were trimmed by using Trimmomatic (v0.38) [7]. After trimming, the remaining high-quality reads were aligned against the *Bos taurus* reference genome assembly ARS-UCD1.2 by using BWA-MEM (0.7.13-r1126) [8]. To obtain highly confident variants, we employed Samtools [9], Picard tools (http://broadinstitute.github.io/picard (accessed on 1 September 2020)), and the Genome Analysis Toolkit (GATK version 3.6–0-g89b7209). Based on the latest reference assembly (ARS-UCD1.2), the SNPs were functionally annotated using ANNOVAR [10].

### 2.5. Analysis of the Population Genetic Structure and Relatedness

The autosomal SNPs were further filtered for missing genotypes and pruning of genotypes with a parameter (—indep-pairwise 50 5 0.2) using PLINK [11]. To calculate linkage disequilibrium (LD) decay, PopLDdecay was used [12]. Additionally, we calculated the inbreeding coefficient (—het) and the nucleotide diversity (π) using VCFtools [13], respectively. The plot, as mentioned above, was depicted using R script (http://www.r-project.org) (accessed on 1 September 2020).

To explore ancestry proportions, we added the genomic data for possible ancestral components and examined the population structures with genetic clusters of K ranging from 2 to 8 using the ADMIXTURE program version 1.3 [14]. We employed MEGA v7.0 to construct an unrooted neighbor-joining tree based on the matrix of the pairwise genetic distance [15]. After construction, it was beautified with iTOL v5 [16]. The smartPCA of the EIGENSOFT v5.0 package was used to perform the principal component analysis (PCA) [17].

### 2.6. Local Ancestry Inference

To infer the haplotype phase and impute the missing allele, we used Beagle v4.1 [18] with default parameters. Local ancestry information was inferred using the software package RFMix in the Dehong humped cattle [19]. We selected Chinese indicine and Indian indicine as reference panels and performed a chi-square test to compare the number of ancestry-specific haplotypes for all segments (*p*-value < 0.05). Based on the *Bos taurus* reference genome, a custom Perl script was used to annotate the segments.

### 2.7. Detection of Selection Signatures

To identify the selective regions in Dehong humped cattle, the nucleotide diversity (θπ) and composite likelihood ratio (CLR) were performed [20]. Briefly, θπ was run with the parameters in VCFtools (50 kb windows with 20 kb steps). CLR was run (with the parameters of 50 kb windows in SweepFinder2) [21].

The fixation index (*F_ST_*) and cross-population extended haplotype homozygosity (XP-EHH) have effective tools for detecting select elimination regions when strong selection signals are obtained. We separately calculated the *F_ST_* and XP-EHH among the two cattle breeds using VCFtools (50 kb windows with 20 kb steps) and selscan v1.1 [22]. We chose Hanwoo cattle and Dehong humped cattle due to the genetic separations. To obtain more reliable results, we used four overlapped methods (*p* < 0.01). It is noteworthy that we calculated Tajima’s D statistic to consolidate our results by using VCFtools.

### 2.8. Enrichment Analyses of Candidate Genes

To understand the functions and complex pathways of candidate genes, we used KOBAS 3.0 (http://kobas.cbi.pku.edu.cn/) (accessed on 1 September 2020), including the Kyoto Encyclopedia of Genes and Genomes (KEGG) and Gene Ontology (GO) in the present study (species: cow, corrected *p*-value < 0.05).

## 3. Results

### 3.1. Genome Resequencing, SNP Identification, and Diversity

Each sample was sequenced, which generated clean data ranging from 180,531,582 to 391,083,340 reads. After aligning with the *B. taurus* reference genome (ARS-UCD1.2), the population reached ~99.7% genome coverage at a depth of 10.5X (Appendix A). The data were jointly genotyped with 53 genomes from around the world (Appendix A). In total, the number of common/shared SNPs was 5.3 million across the 5 cattle populations, while the number of population-specific SNPs was 32.2 million in Dehong humped cattle (Figure 1A). It is important to note that a higher number of unique SNPs was found in Dehong humped cattle. That presented the richer genetic diversity of Dehong humped cattle. The vast majority of SNPs were annotated in intergenic (60.8%) and intronic (37.6%) regions. Furthermore, 0.7% of the SNPs were present in Exon, including 144,663 nonsynonymous and 92,411 synonymous SNPs (Appendix A).

The nucleotide diversity of Dehong humped cattle is between that of Chinese indicine and Indian indicine (Figure 1B). The linkage disequilibrium (LD) decay in Chinese indicine was the fastest, followed by Dehong humped cattle and Indian indicine (Figure 1C). The two results above are the same. For the analysis of the genomic characteristics, the genome variations (the inbreeding coefficient) did not show entirely consistent patterns (Figure 1D). The inbreeding coefficient (—het) comparisons among the five populations showed that the Dehong humped had an overall lower level than that of the commercial cattle.

### 3.2. Population Structure and Demographics

We used 71 whole genome sequencing datasets from 5 breeds for an association study. Both the neighbor-joining (NJ) tree (Figure 2A) and the first two principal components (Figure 2B) derived from autosomal SNP data indicated that taurine and indicine formed separate clusters. A more accurate result involved separating the Dehong humped cattle from Chinese and Indian indicine, respectively. At the optimal number K = 2 with the smallest cross-validation error, the two genetic clusters were observed: one for taurine and the other for indicine cattle. Additionally, we noted the genetic ancestry of Dehong humped cattle in Chinese indicine and Indian indicine at K = 3. Interestingly, the results showed that the genetic influence was more pronounced in Indian indicine than in Chinese indicine (Figure 2C).

### 3.3. Local Ancestry Inference of Dehong Humped Cattle

In addition to the ADMIXTURE method, we used RFMix to infer local ancestry information. Similar to the ADMIXTURE method, it was observed that Dehong humped cattle have ancestral contributions from two origins. To separately identify segments with higher proportions of these two ancestors than the genome-wide proportion, our data were analyzed using the chi-square test for all segments. Ultimately, 444 Chinese indicine and 703 Indian indicine segments were retained (*p* < 0.05) (Figure 3A) (Appendix A). A total of 216 genes were found in the Chinese indicine segment of the Dehong humped cattle. The functional enrichment analysis of these genes included the KEGG pathway and GO terms (corrected *p*-value < 0.05) (Figure 3B). The enrichment analysis revealed that the significant pathway was the actin cytoskeleton organization (GO:0030036, corrected *p*-value = 0.0000247). The genes were selected (*WASF2* [23], *EHBP1* [24], *SPECC1, PFN1* [25,26], *SSH2* [27], *ARHGAP26,* and *DLC1* [28]), which were related to growth traits and the body shape. Thus, we can speculate that the body sizes of Dehong humped cattle may be related to Chinese indicine. Other significant pathways were Hippo signaling (bta04390, corrected *p*-value = 0.025903318), receptor localization to synapse (GO:0097120, corrected *p*-value = 0.0000543), acylglycerol lipase activity (GO:0047372, corrected *p*-value = 0.025903318), and negative regulation of transcription by RNA polymerase II (GO:0000122, corrected *p*-value = 0.000188112) (Appendix A). In addition, 296 genes were annotated in the Indian indicine segment of Dehong humped cattle. As above (Figure 3C), the most pathways were involved in the regulation of inflammatory response (GO:0050727, corrected *p*-value = 0.0177797870524) and B cell homeostasis (GO:0001782, corrected *p*-value = 0.0463632673798). Genes in inflammatory pathways are also associated with the immune response (*ALOX15* [29], *ESR1* [30], *TMEM173* [31], *LYN* [32], *AKNA* [33] and *RC3H1* [34], *LYN* [35], and *NCKAP1L* [36]). The results suggest that Indian indicine could contribute to immunity in Dehong humped cattle. Other important pathways were verified, such as morphine addiction (bta05032, corrected *p*-value = 0.000305455728886), the phospholipase D signaling pathway (bta04072, corrected *p*-value = 0.0463632673798), glutamate receptor activity (GO:0008066, corrected *p*-value = 0.0393954758617), and regulation of long-term neuronal synaptic plasticity (GO:0048169, corrected *p*-value = 0.0447037951668) (Appendix A).

### 3.4. Patterns of Selection

By using two different methods for the signature of selection, we found 539 (composite likelihood ratio, CLR) (Appendix A) and 2297 (nucleotide diversity, θπ) (Appendix A) candidate genes in Dehong humped cattle (Figure 4A). Among these, 428 genes were overlapped, which were considered to be candidate genes and enriched using GO annotation and KEGG pathway terms. The results represented significant enrichment of 2 KEGG pathway terms and 15 GO terms (Appendix A).

To further understand the underlying genetic mechanisms in Dehong humped cattle, we implemented two methods (*F_ST_* and XP-EHH) to detect the positive selection characteristics by comparing the differences in ecological adaptability between Dehong humped and Hanwoo cattle (Figure 4A) (Appendix A). It is worth noting that 82 genes were detected among the four methods mentioned above, indicating that these genes were strongly selected in Dehong humped cattle (Appendix A). The annotations of candidate genes revealed the functions that may be associated with heat tolerance (*FILIP1L* and *ABHD6*) [37,38]. Interestingly, *ABHD6* enables acylglycerol lipase activity, which is a negative modulator of adaptive thermogenesis [38]. Moreover, Tajima’s D and haplotype patterns were used to further validate the selection of the *ABHD6* gene in Dehong humped cattle (Figure 4B,C). Two missense mutations in *ABHD6* showed distinct allelic patterns in Dehong humped cattle (c.870C > A *p*.Asp290Glu; c.987C > A p.Ser329Arg). In addition, we obtained genes (*GZMM*, *PRKCZ*, *STOML2*, *LRBA*, and *PIK3CD*) related to the immune response [39,40,41,42,43].

## 4. Discussion

Population genetic diversity is an important basis for safeguarding the evolution of species [44,45]. It is generally accepted that the higher the genetic diversity of the species, the more adaptable its offspring will be (so expansion is more likely to occur). In addition, the assessment of genetic diversity and the inbreeding coefficient of the population is essential for the use and conservation of the breed’s genetic resources. Nucleotide diversity is the most important factor affecting genetic diversity. In this study, the nucleotide diversity was highest in Chinese indicine, which may be explained by population expansion or introgression [6]. The nucleotide diversity of Dehong humped cattle is second only to that of Chinese indicine, with a wealth of genetic information obtained from two genetic resources (Chinese and Indian indicine). The monotonous environment and high degree of artificial breeding may be the reasons for the lowest nucleotide diversity of Yanbian and Hanwoo cattle [46,47]. This is from the higher inbreeding coefficient and faster LD decay. It also showed the low level of selection and development potential concerning Dehong humped cattle. Effective measures to conserve genetic diversity are conducive to the development of the genetic resources of Dehong humped cattle.

Studies on population structures and phylogenetic relationships are of great importance for understanding historical demographic patterns and tracing ancestral information. Dehong humped cattle consist mainly of Chinese and Indian indicine, which are inextricably linked to the human and geographical environments of their habitats. The home tract of Dehong humped cattle at the border between China and Myanmar (and separated from the rest of China by the Salween River to the east) possesses a unique pedigree composition among Chinese native cattle. Thus, the pedigree compositions of Dehong humped cattle provided the context for exploiting the economic effects of the breed.

To reveal the ancestral proportions of Dehong humped cattle in the local region, the application of RFMix was beneficial to infer local ancestry information. Based on the Kobas 3.0 annotation, the influences of ancestral segments were explored in Dehong humped cattle and we found that Dehong humped cattle received ancestral contributions from Indian indicine and Chinese indicine. Of these, the body sizes of the Dehong humped cattle are influenced by the Chinese indicine segments. The excessive segments inherited from Chinese indicine contained *WASF2* and *PFN1*, two genes belonging to GO terms—actin cytoskeleton organization. The *WASF2* gene was a downstream effector molecule that involved signal transductions from tyrosine kinase receptors and small GTPases to the actin cytoskeleton. promoting actin filament formation [48,49]. The gene can respond to changes in the external environment by reregulating its expression or distribution to further influence the cytoskeleton arrangement [50]. *PFN1* plays an important role in actin dynamics by regulating actin polymerization in response to extracellular signals. Previous studies have shown that *PFN1* could contribute to essential roles in postnatal skeletal homeostasis [51]. Among excessive segments inherited from Indian indicine, the following genes were found to be annotated: *LYN* and *NCKAP1L*. *LYN* plays an important role in the regulation of innate and adaptive immune responses, hematopoiesis, responses to growth factors and cytokines, integrin signaling, as well as responses to DNA damage and genotoxic agents [32]. *NCKAP1L* controls lymphocyte development, activation, proliferation and homeostasis, erythrocyte membrane stability, as well as phagocytosis and migration by neutrophils and macrophages [52,53]. Therefore, the additional ancestral segments (Indian indicine) acted on the immune system of Dehong humped cattle. However, this is just a conjecture, and more theoretical and experimental support is required for further elaboration.

Dehong humped cattle, due to the humid and hot living conditions of their home tracts, have remarkable heat-tolerant properties. The gene *ABHD6*, selected by four methods, has been associated with heat tolerance in Dehong humped cattle. *ABHD6* is located on chromosome 22 of cattle at about 0.048 Mbp. This region showed extreme differentiation and distinct haplotype patterns in two populations (Dehong humped and Hanwoo cattle). Tajima’s D analysis exhibited a significantly lower value in Dehong humped cattle. The results verified that the *ABHD6* gene, which is associated with heat tolerance, showed strong positive selection in Dehong humped cattle. Additionally, missense mutations in *ABHD6* may have crucial effects on heat resistance in Dehong humped cattle. In addition to genes related to heat tolerance, we obtained genes related to immune response, such as *GZMM*, *PRKCZ*, *STOML2*, *LRBA*, and *PIK3CD. LRBA* contributes to the secretion of the immune effector molecules. The immune genes in Dehong humped cattle may make them more able to cope with pathogenic challenges in the local environment.

## 5. Conclusions

In this study, WGS data were used to investigate the population structure of Dehong humped cattle, leading to the first in-depth research on gene diversity, phylogenetic relationships, ancestry components, and genomic regions under selection. The identified genes will be helpful to better understand the features of Dehong humped cattle and further unravel the characteristics of other native cattle in China. The revelation of the genetic diversity of Dehong humped cattle will establish a sound foundation for conservation and breeding programs in the future.

## Figures and Tables

**Figure 1 biology-11-01331-f001:**
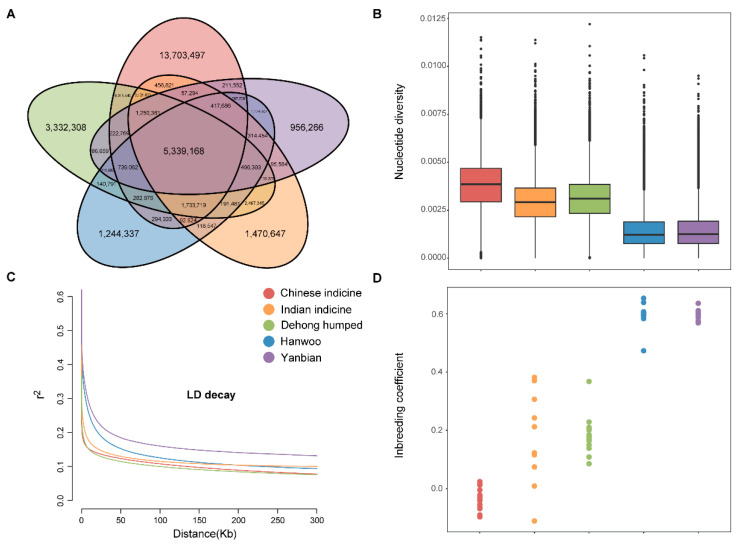
Genetic diversity among 71 samples from 5 populations. (**A**) The numbers of shared SNPs and private SNPs across the five populations by the Venn diagram. (**B**) The nucleotide diversity for each group by box plots. (**C**) Decay of linkage disequilibrium on cattle autosomes estimated from each breed. (**D**) Inbreeding coefficient for each individual.

**Figure 2 biology-11-01331-f002:**
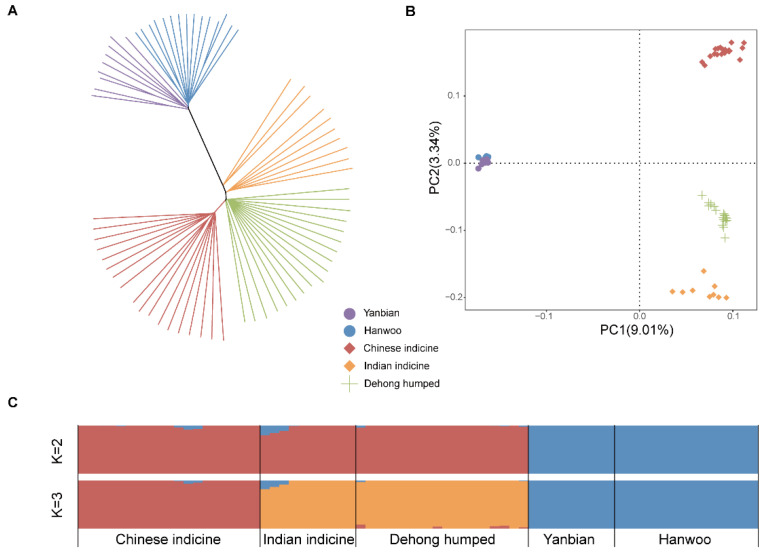
Population genetic analysis. (**A**) The neighbor-joining tree of cattle. (**B**) The principal component analysis of cattle with PC1 (9.01%) vs. PC2 (3.34%). (**C**) Genetic structure of cattle using ADMIXTURE with K = 2 and K = 3.

**Figure 3 biology-11-01331-f003:**
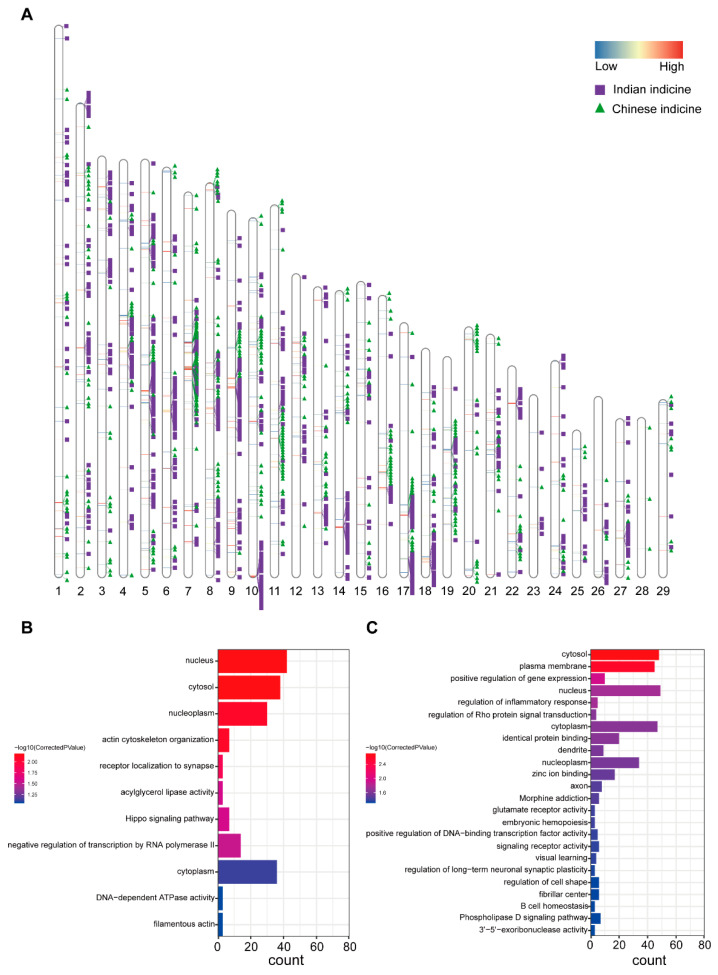
Identification of the local segments in which proportions of a certain ancestry were significantly higher than the proportion in the whole genome in Dehong humped cattle. (**A**) Distribution of the local segments whose proportions of Chinese indicine and Indian indicine were excessive compared with the average level of the whole genome. (**B**) The KEGG pathways and gene ontology from the enrichment analyses of genes with excessive Chinese indicine proportions. (**C**) The KEGG pathways and gene ontology from the enrichment analyses of genes with excessive Indian indicine proportions.

**Figure 4 biology-11-01331-f004:**
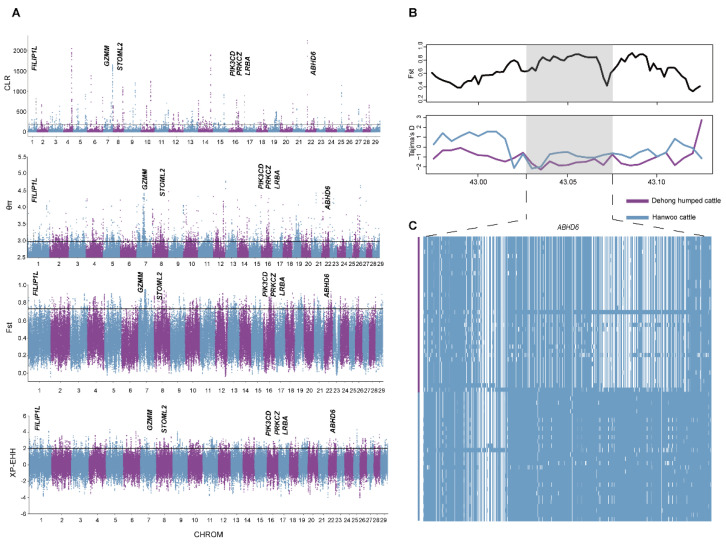
The signatures of the positive selection in Dehong humped cattle. (**A**) Manhattan plot of selective sweeps by θπ, CLR, *F_ST_*, and XP-EHH. (**B**) *F_ST_* and Tajima’s D plots of the *ABHD6* gene. (**C**) Haplotype diversity of the *ABHD6* gene.

## Data Availability

Sequence data were deposited in GenBank (BioProject accession number: PRJNA780661).

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
