# Peer review of "Assessing the Role of Ancestral Fragments and Selection Signatures by Whole-Genome Scanning in Dehong Humped Cattle at the China–Myanmar Border"

_biology, 2022, doi:10.3390/biology11091331_

Round 1

Reviewer 1 Report

The study is mainly important for the understating of the genetic diversity and genomic regions conserved in the Dehong humped cattle population raised in China. However, the introduction is short and without much detail about the selection history of this breed. As this is a specific Chinese cattle population with relevance in Asia, it is difficult for outside readers to know the breed’s history. It would be important to provide more details in the Introduction section.

The authors commented on the establishment of genetic conservation and breeding programs for the breed, but it is not possible to know the current breed situation or if the breed is being used for selection or conservation purposes. Based on the results and comments of the authors, the breed is the one with the greatest genetic diversity among those breeds analyzed, with no apparent risks. I recommend commenting more on the breed's situation in the introduction.

Regarding the methodology, the authors performed robust analyses that provided varied results, but it seems that the results have not been explored much and discussed well, which opens up several open questions about the genetic diversity and inbreeding in the Dehong humped cattle breed. Also, the authors used different approaches to identify signatures of selection but the importance of the chosen methods and differences among them or reasons for the choice of such methods does not appear in the text. These points or more on the signature of selection outcomes should be better discussed in the manuscript.

More points should deserve the authors’ attention. There were 10 counts of indicine animals used in the comparisons, with breeds with one or two representants. If we assess the literature on the population structure of indicine breeds, we will see they are very divergent populations. What is the implication of this low number of animals used in the comparisons for the study? Should it cause biased results and interpretations?

Could the number of animals provide reliable results on the genetic diversity among the breeds? The author should support their findings with studies from the literature.

It is not clear what inbreeding coefficient was used. Specify that in the M&M and make sure to mention the method when discussing the results.

Find more comments below.

Line 43: B. indicus in italic.

Check citation for ‘Yunnan 50 Commission of Animal Genetic Resources 2014)’.

Line 54: what do the authors mean by “stable genetic characteristics”?

Line 69: Include the process number in the Ethics statement item.

Line 144: Bos taurus in italic

Lines 121-123: missing reference for this phrase.

Line 125: why four methods? I see five methods.

Line 128: Specify the database and parameters used in the enrichment analyses.

Line 160: I am not sure if ‘association study’ is the best choice of term to be used. This sentence is followed by an incomplete sentence.

Line 182: ‘the most pathway was’. Rewrite the sentence.

Line 195: ‘these pathways’. What pathways? The authors should specify the pathways or improve the quality of the sentence.

In section ‘3.3. Local ancestry inference of Dehong humped cattle’, it would be more interesting if the authors had raised a concern on the regions conserved in the Dehong humped cattle specifying the identified chr and segment (start and end) instead of the pathways. Those results regarding chr and position can be useful for future studies. The main regions can appear in a table.

Regarding the methodology for the RFMix analyses, what are the implications of using Bos taurus as the reference genome? Can that influence the results obtained in the study designed by the authors? Provide detail on the method used in RFMix in the M&M section or discuss that in the discussion section.

Lines 219-222: Why did the authors choose Hanwoo? I am not sure if that was mentioned in the manuscript.

Line 227: typo in ‘nagative’.

Paragraph in lines 219-231: Specify in the text what result comes from Tajima’s D method.

Lines 238-239: include reference for that statement.

Line 257: “Thus, our research provided the context for exploiting the economic effects of the breed.” How these results can contribute to exploiting the economic effects of the breed?

Line 260: Verify the word ‘aoolication’.

Line 284: text of hard comprehension or missing words. Rewrite the sentence “All four methods selected for the gene 284 ABHD6…”

Line 288: text of hard comprehension or missing words. Rewrite the sentence.

Lines 292-294: The word study appears three times in the sentence. Rephrase the sentence.

Line 294: What do the authors mean by Selective Sweep Test? Signatures of selection approaches?

Line 260: RFmix or RFMix? Use a standard form along the manuscript.

Lines 266, 287: Avoid using parentheses. Try to describe the detail or provide an explanation. ‘GO terms (actin cytoskeleton organization)’.

Line 288: Tajima’D or Tajima’s D?  Rewrite the sentence.

Line 236: A short discussion was provided by the authors as a whole, especially for the signatures of selection results.

Reviewer 2 Report

The present study has a niche character, being useful to a limited number of researchers. The implications of this study are of interest only to specialists in the field, the directly interested farmers not having the opportunity to implement the results of this study in current practice. Anyway, congratulations on the work done.

Author Response

Thank you for your advice.

Reviewer 3 Report

I would like to thank the Authors and Editors of the Manuscript "Assessing the role of ancestral fragments and selection signatures by Whole-genome scanning in Dehong humped cattle at China-Myanmar border".

To my understanding, this study compares Dehong humped cattle with four other local cattle breeds, reconstructing population structure and detecting signatures of diversification and selection, contextualized in terms of ancestry derived from other breeds (specifically, Chinese indicine and Indian indicine). Gene ontology and enrichment analysis reveal traits that may have been adapted in Dehong cattle for immune response and heat tolerance.

The overall study is well performed, with clear methodology, interesting results and a sound discussion paragraph which is supported by the discoveries presented in the manuscript.

I would appreciate, however, a relatively extended introduction, especially regarding the relationships among the five chosen cattle breeds and a justification as to why they have been specifically selected for the study (so, why using two Korean cattle breeds to study cattle diversification and genetic peculiarities at the China-Myanmar border?).

I would also suggest to report a graph of the CV errors, as well as a Figure of all the runs of Admixture performed (for K=2 to 8) in the Supplementary Materials.

There are several sentences that are relatively unclear in terms of English grammar used, so I would suggest an extended review of English language.

Other than that, I really appreciate the intent and content of this work.

Reviewer 4 Report

1.      Line 243- Nucleotide diversity cannot be sole determinant of genetic diversity. Factors affecting effects of nucleotide diversity into functionality such as how much of this nucleotide diversity is translated into protein synthesis, any potential regulators which would determine functionality of diversity need to be explained.   

2.      Line 255-256- Environmental effect of this geographical isolation/ uniqueness such as effect of feed and water availability to cattle and their impact on phenotypical features can be discussed.

3.      Line 288- what is significantly lower in the cattle, please elaborate.

4.      Line 297- how identification genetic diversity would shape breeding program, kindly explain.

Reviewer 5 Report

The manuscript provides some useful information about the signature of selection for Dehong cattle.  Overall, the approaches are fine, and the methods are suitable for the analyses. The sample size is relatively small. The authors might provide some more details on the methods as well as some justification for the choices of the parameters. 

Line 14: change:  other native cattle to another native cattle

Line 15: “  It is speculated how” is not a clear term

Line 16: Selection changes to the selection

Line 18: Names the traits for  economically important traits

Line 19: remove sound or replace it with a more specific word

Line 22: Might add sequencing methods and coverage

Line 23: How many animals from publicly-available data?

Line 25: it is not very popular for the RFmix method if possible extend a phrase about this approach

How many variants are used for the signature of the selection

Line 56-57: How many samples did these authors use?

Line 76: it is not clear (including  data downloaded from five).

Why did the authors choose to download these animals, are they all animals having the sequences in NCBI?

Are there any differences in the sequencing methods used to obtain the sequence for these downloaded sequences?

Line 89-95: More details about the parameters for each method should be provided

Line 96: did the authors perform the quality control for SNPs, how about indels and other variant types?

Line 119: please justification for this selection “50kb sliding window and 20kb step in VCFtools.”

Line 199: Which methods for correcting P values?

The authors do not need to add many digitals corrected p values, just two or three enough

For the methods, did authors able to check how many SNPs in the current study overlapped with the HD Chip, it is important

Line294: why capitalizing “Selective Sweep Test.”

Round 2

Reviewer 1 Report

The authors considered my suggestions in the first review round. I have some other comments.

Abstract

data of (n=18) Dehong humped cattle à data of Dehong humped animals (n=18)

Introduction

“stable genetic characteristics” and “free from the infection of foreign commercial cattle”. I understood the authors comments but it is better to change the text. Provide the same explanation found in the author's reply file and use technical terms. I believe infection is not the most appropriate term.

Section 2.5: Provide the reference for PLINK.

Section 2.6: “In order to obtain more reliable results, we used four overlapped methods (p < 0.01).” From the comments in the author’s letter, it is important to clarify the idea. It does not make sense to use a p-value since there are no comparison and statistics. Suggestion:  We used four different methods to verify common regions of selection signal.

Section 3.2: New inclusion: Correct the text (“are formed separate clusters”).

Section 3.3:

Lines 184 and 185: “two ancestors”. There is a better way to mention the idea. Suggestion: ancestral contribution from two origins, Indian indicine and Chinese indicine?

Line 192: “The genes were performed”. Correct the text.

Lines 192 and 204: Use comma in the text instead of |.

Line 206: There are many other important pathways… -> Other important pathways were verified…

Line 267: “Thus, our research provided the context for exploiting the economic effects of the breed.” I read the author’s comment. It is important to provide the explanation within the manuscript, otherwise your readers won’t have access to that.

Line 288: NCKAP1L Controls à NCKAP1L controls

Lines 273-275: “Of these, the body size of the Dehong humped cattle is influenced by the Chinese indicine segments, which explained why its body size is slightly smaller than that of the Indian indicine.” There is few evidence in the results to make this association and body size is a complex trait influenced by many variants. Perhaps the author should be careful in making this statement. The common segments can be associated with other phenotypic similarities among animals from the two populations.

Lines 300-301: “The result suggested that the haplotype patterns and Tajima’s D showed strong positive selection in Dehong humped cattle”. You are in the discussion section; we should expect the authors provide an explanation for that and not only introduce again the result. Here is my question: what does this result mean? And my suggestion: include the comments within the manuscript.

Conclusion

Line 310: “selective sweep test”. I will insist in this term again. Selective sweep gives an idea of analysis based on LD and haplotype. The authors should consider a term that will incorporate the other performed analysis, for instance Fst was also used and is based on allele frequency.

My other comment is to avoid the use of citation in the conclusion section -> Exclude “[53]”

Author Response

Thank you for your advice. We have revised and resubmitted the manuscript in accordance with the comments.